# Sensitivity Analysis for Survival Prognostic Prediction with Gene Selection: A Copula Method for Dependent Censoring

**DOI:** 10.3390/biomedicines11030797

**Published:** 2023-03-06

**Authors:** Chih-Tung Yeh, Gen-Yih Liao, Takeshi Emura

**Affiliations:** 1Department of Information Management, Chang Gung University, Taoyuan 33302, Taiwan; 2Biostatistics Center, Kurume University, Kurume 830-0011, Japan; 3Research Center for Medical and Health Data Science, The Institute of Statistical Mathematics, Tokyo 190-8562, Japan

**Keywords:** copula, Cox regression, dependent censoring, gene expression, high-dimensional data, Kendall’s tau, lung cancer, prognostic prediction, survival analysis, survival prediction

## Abstract

Prognostic analysis for patient survival often employs gene expressions obtained from high-throughput screening for tumor tissues from patients. When dealing with survival data, a dependent censoring phenomenon arises, and thus the traditional Cox model may not correctly identify the effect of each gene. A copula-based gene selection model can effectively adjust for dependent censoring, yielding a multi-gene predictor for survival prognosis. However, methods to assess the impact of various types of dependent censoring on the multi-gene predictor have not been developed. In this article, we propose a sensitivity analysis method using the copula-graphic estimator under dependent censoring, and implement relevant methods in the R package “compound.Cox”. The purpose of the proposed method is to investigate the sensitivity of the multi-gene predictor to a variety of dependent censoring mechanisms. In order to make the proposed sensitivity analysis practical, we develop a web application. We apply the proposed method and the web application to a lung cancer dataset. We provide a template file so that developers can modify the template to establish their own web applications.

## 1. Introduction

Prognostic analysis for survival often employs gene expressions obtained from high-throughput screening for tumor tissues from patients [1,2,3,4,5,6]. Here, the primary interest is often to find a set of genes that are strong predictors of survival. The simplest and yet most commonly used approach is to select a small number of significant genes found in a sequence of univariate Cox models fitted to patients’ survival data [5,6,7,8,9].

A linear predictor using multiple genes has been shown to be useful for survival prediction. Usually, a multi-gene predictor is more accurate than a single-gene predictor as the prognostic ability of a single gene is limited. A predictor optimally constructed from the selected genes becomes a useful biomarker for predicting survival in breast cancer [10,11,12,13,14,15], lung cancer [6,7,8,9,16,17], gastric cancer [18,19], ovarian cancer [20,21,22,23,24], skin cancer [25], liver cancer [26,27], bladder cancer [28], head and neck cancer [29,30], glioma [31], myeloproliferative neoplasms [32], kidney cancer [33], and cancers of mixed types [34,35]. These analyses were performed mostly based on univariate Cox regression with the significance scaling of *p*-values, such as 0.05, 0.01, and 0.001, followed by cross-validation and/or external validation. The primary reason for applying Cox regression is that it can easily handle censored survival times, i.e., survival time may not be observable for all patients under investigation.

However, Cox regression critically relies on the *independent censoring assumption*: survival time and censoring time need to be statistically independent. This assumption is easily violated when patients drop out from the study due to the worsening of his/her health condition or patients are removed for transplantation [36,37]. Dependent censoring also arises due to unobserved factor [38,39,40,41]. In such cases, survival time is censored dependently on their health status. If the independent censoring assumption is violated, Cox regression analyses may not correctly identify the effect of each gene and thus may fail to select truly effective genes [41]. Therefore, the resultant predictor based on selected genes may have reduced ability to predict survival.

A copula-based gene selection approach [41] can effectively adjust for dependent censoring, yielding a multi-gene predictor. Besides the development of a multi-gene predictor, copula-based dependent censoring models have gained popularity in recent years for dealing with survival data [42,43,44,45,46,47,48,49,50]. However, methods to assess the impact of various types of dependent censoring on the multi-gene predictor have not been developed. In this article, we propose a sensitivity analysis method using the copula-graphic estimator [47,51,52,53,54,55,56] under dependent censoring, and implement relevant methods in the R package *compound.Cox* [9]. While the CG estimator has mostly been discussed for estimation of survival functions, its adaptation to the context of survival prognostic prediction remains unclear. This article aims to fill this gap by proposing detailed statistical methodologies; see Section 3 for details.

Another important goal of this article is to establish a Shiny-based web application (in Section 4) that can be manipulated by any user online without installing any software. This effectively connects the development of a sensitivity analysis and its implementation in order to carry out a patient’s prognosis. The application is written by R packages *Shiny* [57] and *compound.Cox* [9]. While this application is developed using data on lung cancer patients, the computational framework can be extended to other survival data from other patients. We provide the R code in Appendix A so that users can easily edit the code to be adapted to their own data analyses.

The article is constructed as follows: Section 2 reviews the backgrounds of this research. Section 3 introduces the proposed sensitivity analysis method. Section 4 describes the development of the software and web application. Section 5 provides the results of applying the proposed method to the lung cancer data. Section 6 concludes with discussion.

## 2. Backgrounds

In this section, we shall first review gene selection methods [1,9,41] based on survival data. Gene selection is a required step before building a prognostic prediction method. Next, we shall review prognostic prediction methods based on selected genes.

We define basic notations as follows: $T_{i}$ is survival time and $x_{i}={\left(x_{i1},\;\ldots ,\;x_{ip}\right)}^{’}$ is a p-vector of genes from individuals i=1, 2,…, n. The conditional survival function given $x_{i}$ is denoted by $S\left(t|x_{i}\right)=P\left(T_{i}>t|x_{i}\right)$, and the conditional hazard function given $x_{i}$ is defined by $h\left(t|x_{i}\right)=\left[-dS\left(t|x_{i}\right)/dt\right]/S\left(t|x_{i}\right)$. Similarly, $h(t|x_{ij})$ denotes the conditional hazard function given the *j*-th gene $x_{ij}$.

### 2.1. Classical Gene Selection Method

The approach called *univariate selection* [1,9] is performed as follows: In the initial step, a significance of each gene is examined by univariate Cox regression one-by-one. Then, a subset of significant genes is selected with a *p*-value threshold, such as 0.05, 0.01, and 0.001. This is the most standard approach for gene selection in medical research.

In the univariate selection, a significance of gene $x_{ij}$ on survival time $T_{i}$ is measured by fitting a Cox proportional hazards model [58] given by
(1)$$h\left(t|x_{ij}\right)=h_{0j}\left(t\right)e^{\beta_{j}x_{ij}},\;\;\;\;\;\;j=1,\;\ldots ,\;p$$
for each gene j. Here, the parameter $\beta_{j}$ measures the association between survival and the gene, and the baseline hazard function $h_{0j}\left(.\right)$ is a nuisance parameter. Cox’s partial likelihood estimator [58], denoted by ${\hat{\beta }}_{j}$, is used to obtain the *p*-value for the Wald test for $H_{0j}:\beta_{j}=0$ vs. $H_{1j}:\beta_{j}\neq 0$ without specifying $h_{0j}\left(.\right)$. One selects genes that exhibit smaller *p*-values than a threshold value. While the Cox’s approach is simple and widely accepted, the validity of the gene selection method relies on an important assumption.

To describe the assumption for censoring mechanisms, we define $U_{i}$ as random censoring time. The (training) samples consist of $\left(t_{i},\;\delta_{i},\;x_{i}\right)$, i=1, 2,…, n, where $t_{i}=\min\left(T_{i},U_{i}\right)$ and $\delta_{i}=1\left\{T_{i}\leq U_{i}\right\}$, where $1\left\{\cdot \right\}$ is the indicator function. The samples are often referred to as *training samples* since they are used to train the model (1). For the estimator ${\hat{\beta }}_{j}$ to capture the true $\beta_{j}$ in Equation (1), one needs to impose the following assumption [37,41]:

***Independent censoring assumption****: Survival time and censoring time are independent conditionally on* $x_{ij}$*. That is,*$$P(T_{i}>t,\;U_{i}>u|x_{ij})=P(T_{i}>t|x_{ij})\times P(U_{i}>u|x_{ij}),\;\;\;\;\;i=1,\;2,\ldots ,\;n;\;j=1,\;\ldots ,\;p$$

If this assumption is violated, Cox’s estimator ${\hat{\beta }}_{j}$ is biased for the true value of $\beta_{j}$ [37,41]. This assumption is violated when $U_{i}$ is dropout time from the study due to the worsening of his/her health status or $U_{i}$ is time at removal for transplantation [36,37].

### 2.2. Copula-Based Gene Selection Method

Copulas provide a general model to relax the assumption of independent censoring. Copula-based dependent censoring models have gained popularity in recent years for dealing with survival data [41,42,43,44,45,46,47,48,49,50,55,56]. For the purpose of gene selection, [41] introduced the following copula model:(2)$$P(T_{i}>t,\;U_{i}>u|x_{ij})=C_{\alpha }\left(P(T_{i}>t|x_{ij}),P(U_{i}>u|x_{ij})\right)\;,\;\;\;\;\;i=1,\;2,\ldots ,\;n;\;j=1,\ldots ,p$$
where $C_{\alpha }$ is a copula [59] with a parameter α that specifies the strength of dependence.

The copula model (2) can specify a variety of dependent censoring structures. For implementation, a copula with a simple form, such as the Clayton, Gumbel, or Frank copula, is suggested. Furthermore, a copula is better parameterized so that α=0 gives the independence copula $C_{\alpha =0}\left(v,w\right)=vw$. An example is the Clayton copula of the form
$$C_{\alpha }\left(v,w\right)={\left(v^{-\alpha }+w^{-\alpha }-1\right)}^{-\frac{1}{\alpha }},\;0<v,w<1,\;\alpha >0$$

This copula accommodates the independence copula as its limiting case by $C_{\alpha }\left(v,w\right)\to vw$ as α→0. As the value of α departs from zero, the level of dependence increases. To measure the degree of dependence, it is convenient to use Kendall’s tau, defined as $\tau =\alpha /\left(\alpha +2\right)$ under the Clayton copula. The value of τ ranges from independence (τ=0) to perfect positive dependence (τ=1). While the Clayton copula permits only positive dependence, other copulas (e.g., the Frank copula) give negative dependence.

Under the Clayton copula model, a significance of gene $x_{ij}$ on survival time $T_{i}$ is measured by fitting the Cox model (1), namely,
$$P\left(T_{i}>t|x_{ij}\right)=S\left(t|x_{ij}\right)=\exp\left(-\int_{0}^{t}h_{0j}\left(u\right)e^{\beta_{j}x_{ij}}du\right).$$

Here, $\beta_{j}$ measures the association between survival and a gene. The significance of a gene is measured by the estimator proposed by [41], denoted as ${\hat{\beta }}_{j}\left(\alpha \right)$. Accordingly, a subset of significant genes is constructed via a *p*-value threshold for testing $H_{0j}:\beta_{j}=0$ vs. $H_{1j}:\beta_{j}\neq 0$. The value of α can be estimated by $\hat{\alpha }$ that maximizes the cross-validated *c*-index [37,41]. The computation is implemented by the R package, *compound.Cox* [9], and explained by the book [37].

### 2.3. Prognostic Prediction Method

Once genes are selected based on survival data, one often wishes to assess their prognostic ability for future patients. In many cases, the metric of interest is the ability of the genes to separate good and poor prognosis groups. For this purpose, researchers typically use a multi-gene predictor, defined as a weighted sum of gene expressions [1,5,6,7,8,9].

To formulate a multi-gene predictor, suppose that q (<p) selected genes are measured for a future patient for a prognostic prediction purpose, denoted as $\left(x_{1},\ldots ,x_{q}\right)$. Survival prediction can be formulated by the prognostic index { XE “prognostic index:prognostic index”}(PI) defined as
$$\mathrm{PI}\left(\hat{\alpha }\right)={\hat{\beta }}_{1}\left(\hat{\alpha }\right)x_{1}+\;\cdots +\;{\hat{\beta }}_{q}\left(\hat{\alpha }\right)x_{q}.$$

The PI is a weighted sum of genes whose weights reflect the degree of univariate association. A high (low) value of the PI gives a high (low) risk of death. Thus, a patient may be classified into either a low-risk group or a high-risk group using a cut-off value (e.g., the median of the PI values calculated from the training samples).

If α=0 were assumed in $\mathrm{PI}\left(\alpha \right)$, the PI would be the classical predictor based on univariate Cox regression under the independent censoring assumption [1,5,6,7,8,9]. As the value of α is estimated by maximizing the cross-validated *c*-index [37,41], the resultant $\mathrm{PI}\left(\hat{\alpha }\right)$ may have an improved predictive ability over the traditional PI. In medical research, it is the standard to display two Kaplan–Meier (KM) estimates for survival functions calculated for the two groups to see if there is a clear separation between them. However, if dependent censoring exists in test samples, the KM estimators are biased [52]. Therefore, we shall propose a method to examine a sensitivity of the PI to dependent censoring.

## 3. Proposed Methods

In this section, we propose a sensitivity analysis method using the CG estimator to assess the impact of dependent censoring on survival prognostic prediction.

This section deals with *test samples* that are different from the training samples used to select genes and develop the PI (Section 2). In our proposed method, the primary role of the test samples is to examine the sensitivity of the prognostic ability of the PI to dependent censoring. In many medical studies, random multiple splits for training/test sets are applied to develop and validate a PI. However, the goal of the proposed sensitivity analysis is different. A sensitivity analysis is performed to examine the potential effect of dependent censoring that may be present in the test samples.

Define $T_{i}$ as survival time, $U_{i}$ as censoring time, and $\left(x_{i1},\;\ldots ,\;x_{iq}\right)$ as a q-vector of genes from individuals i=1, 2,…, n in the test samples. The test samples consist of $\left(t_{i},\;\delta_{i},\;x_{i}\right)$, i=1, 2,…, n, where $t_{i}=\min\left(T_{i},U_{i}\right)$ and $\delta_{i}=1\left\{T_{i}\leq U_{i}\right\}$, where the sample size n is possibly different from the size of the training samples. Let ${\mathrm{PI}}_{i}\left(\hat{\alpha }\right)={\hat{\beta }}_{1}\left(\hat{\alpha }\right)x_{i1}+\;\cdots +\;{\hat{\beta }}_{q}\left(\hat{\alpha }\right)x_{iq}$ be the PI for the *i*-th patient in the test samples, where ${\hat{\beta }}_{j}\left(\hat{\alpha }\right)$ is computed by the training samples (Section 2.2).

Figure 1 shows how the PI is applied to classify patients into a high-risk (low-risk) group if his/her PI is greater (less) than a cut-off value c. Concretely, $\left\{i;{\mathrm{PI}}_{i}\left(\hat{\alpha }\right)\leq c\right\}$ is a good prognosis (low-risk) group, and $\left\{i;{\mathrm{PI}}_{i}\left(\hat{\alpha }\right)>c\right\}$ is a poor prognosis (high-risk) group. If the classification is successful, the conditional survival function $S_{\mathrm{Good}}\left(t|c\right)=P\left(T_{i}>t{|\mathrm{PI}}_{i}\left(\hat{\alpha }\right)\leq c\right)$ is higher than the conditional survival function $S_{\mathrm{Poor}}(t|c)=P\left(T_{i}>t{|\mathrm{PI}}_{i}\left(\hat{\alpha }\right)>c\right)$, namely, $S_{\mathrm{Good}}\left(t|c\right)>S_{\mathrm{Poor}}(t|c)$.

The cut-off value c may be chosen to be the 50th percentile (median) of the PIs that are computed by the training samples. The median was chosen to ensure a desirable statistical power to detect the difference of the two groups and to achieve robust groupings by eliminating the instability due to unevenly allocated sample sizes. Additionally, this rule has been commonly employed in the analysis of survival prognostic prediction with a multi-gene predictor [6,20,25,26].

### 3.1. Sensitivity Analysis via the Copula-Graphic Estimator

This section describes the proposed method of assessing the sensitivity of the PI by using the CG estimator.

Let $\left\{i;{\mathrm{PI}}_{i}\left(\hat{\alpha }\right)\leq c\right\}$ be a good prognosis (low-risk) group, and $\left\{i;{\mathrm{PI}}_{i}\left(\hat{\alpha }\right)>c\right\}$ be a poor prognosis (high-risk) group, where c is a cut-off value. Let $n_{\mathrm{Good}}=\sum_{i=1}^{n}1\left\{{\mathrm{PI}}_{i}\left(\hat{\alpha }\right)\leq c\right\}$ and $n_{\mathrm{Poor}}=\sum_{i=1}^{n}1\{{\mathrm{PI}}_{i}\left(\hat{\alpha }\right)>c\}$ be the sizes of the groups. To assess the sensitivity of the prognostic ability of the PI, one needs to understand how well the two survival probabilities are differentiated between the two groups (Figure 1). Accordingly, one needs to estimate two conditional survival functions:$$S_{\mathrm{Good}}(t|c)=P\left(T_{i}>t{|\mathrm{PI}}_{i}\left(\hat{\alpha }\right)\leq c\right)$$
$$S_{\mathrm{Poor}}(t|c)=P\left(T_{i}>t{|\mathrm{PI}}_{i}\left(\hat{\alpha }\right)>c\right)$$

We suggest estimating them by the CG estimator of [52] that adjusts for the effect of dependent censoring { XE “Kendall’s tau:Kendall’s tau”}. In order to apply the CG estimator, it is necessary to impose the following Archimedean copula models:(3)$$P\left(T_{i}>t,U_{i}>u{|\mathrm{PI}}_{i}\left(\hat{\alpha }\right)\leq c\right)=\phi_{\alpha }^{-1}\left[\phi_{\alpha }\left\{S_{\mathrm{Good}}\left(t|c\right)\right\}+\phi_{\alpha }\left\{P\left(U_{i}>u{|\mathrm{PI}}_{i}\left(\hat{\alpha }\right)\leq c\right)\right\}\right],$$
(4)$$P(T_{i}>t,U_{i}>u|{\mathrm{PI}}_{i}\left(\hat{\alpha }\right)>c)=\phi_{\alpha }^{-1}\left[\phi_{\alpha }\{S_{\mathrm{Poor}}(t|c)\}+\phi_{\alpha }\{P(U_{i}>u|{\mathrm{PI}}_{i}\left(\hat{\alpha }\right)>c)\}\right]$$
where $\phi_{\alpha }$ is a generator function that specifies the dependence structure. It is a continuous and decreasing function such that $\phi_{\alpha }\left(0\right)=\infty$ and $\phi_{\alpha }\left(1\right)=0$ [59]. The value of α in Equations (3) and (4) can be chosen arbitrarily and can be different from the value $\hat{\alpha }$ estimated by the training samples.

If the models (3) and (4) are assumed, one can estimate $S_{\mathrm{Good}}(t|c)$ and $S_{\mathrm{Poor}}(t|c)$ consistently by the following CG estimators:$${\hat{S}}_{\mathrm{Good}}^{\mathrm{CG}}\left(t|c,\alpha \right)=\phi_{\alpha }^{-1}\left[\underset{i:t_{i}\leq t,\;\delta_{i}=1,{\;\mathrm{PI}}_{i}\leq c}{\sum }\left\{\phi_{\alpha }\left(\frac{{\bar{Y}}_{\mathrm{Good}}\left(t_{i}\right)-1}{n_{\mathrm{Good}}}\right)-\phi_{\alpha }\left(\frac{{\overset{‾}{Y}}_{\mathrm{Good}}\left(t_{i}\right)}{n_{\mathrm{Good}}}\right)\right\}\right],$$
$${\hat{S}}_{\mathrm{Poor}}^{\mathrm{CG}}\left(t|c,\alpha \right)=\phi_{\alpha }^{-1}\left[\underset{i:t_{i}\leq t,\;\delta_{i}=1,{\;\mathrm{PI}}_{i}>c}{\sum }\left\{\phi_{\alpha }\left(\frac{{\overset{‾}{Y}}_{\mathrm{Poor}}\left(t_{i}\right)-1}{n_{\mathrm{Poor}}}\right)-\phi_{\alpha }\left(\frac{{\overset{‾}{Y}}_{\mathrm{Poor}}\left(t_{i}\right)}{n_{\mathrm{Poor}}}\right)\right\}\right],$$
where ${\bar{Y}}_{\mathrm{Good}}\left(u\right)=\sum_{i=1}^{n}1\left\{t_{i}\geq u,{\;\mathrm{PI}}_{i}\leq c\right\}$ and ${\bar{Y}}_{\mathrm{Poor}}\left(u\right)=\sum_{i=1}^{n}1\left\{t_{i}\geq u,{\;\mathrm{PI}}_{i}>c\right\}$ are the numbers at risk at u.

Note that the CG estimator reduces to the KM estimator by $\phi_{0}\left(t\right)=-\log\left(t\right)$, and hence, the resultant estimator reduces to the KM estimator, e.g.,
$${\hat{S}}_{\mathrm{Good}}^{\mathrm{KM}}\left(t|c\right)=\underset{i:t_{i}\leq t,\;\delta_{j}=1,{\;\mathrm{PI}}_{i}\leq c}{\prod }\left[1-\frac{1}{{\overset{‾}{Y}}_{\mathrm{Good}}\left(t_{i}\right)}\right].$$

For instance, for the Clayton copula, one has $\phi_{\alpha }\left(t\right)=\left(t^{-\alpha }-1\right)/\alpha \to -\log\left(t\right)$ with α→0. The copula generated by $\phi_{0}\left(t\right)=-\log\left(t\right)$ is the independence copula:$$\phi_{0}^{-1}\left[\phi_{0}\left(v\right)+\phi_{0}\left(w\right)\}\right]=vw.$$

This means that the KM estimator is the CG estimator under the independent censoring assumption.

To compute the CG estimators, a parametric form of the copula must be specified (examples of parametric copulas will be shown in Section 3.2). That is, both the form of $\phi_{\alpha }\left(.\right)$ and the parameter value of α must be specified. However, both of them are difficult to estimate with test samples since they provide modest information about the copula [51,52,60].

Our strategy to tackle this difficulty is a sensitivity analysis that is performed under a variety of copula models [52,53,54,55,56]. More precisely, we suggest plotting ${\hat{S}}_{\mathrm{Good}}^{\mathrm{CG}}(t|c,\alpha )$ and ${\hat{S}}_{\mathrm{Poor}}^{\mathrm{CG}}(t|c,\alpha )$ for the three forms of $\phi_{\alpha }\left(.\right)$ and many different parameter values of α. If ${\hat{S}}_{\mathrm{Good}}^{\mathrm{CG}}(t|c,\alpha )$ and ${\hat{S}}_{\mathrm{Poor}}^{\mathrm{CG}}(t|c,\alpha )$ are clearly separated for a variety of $\phi_{\alpha }\left(.\right)$ and α, we conclude that the prognostic ability the PI is shown to be robust against dependent censoring. This strategy is in line with previous sensitivity analyses [52,53,54,55,56], yet it is a new method in the context of prognostic survival analysis.

An objective metric of “clear separation” between ${\hat{S}}_{\mathrm{Good}}^{\mathrm{CG}}(t|c,\alpha )$ and ${\hat{S}}_{\mathrm{Poor}}^{\mathrm{CG}}(t|c,\alpha )$ is obtained via a significance test proposed by Emura and Chen [37,41]. Formally, one can reject the null hypothesis $H_{0}:S_{\mathrm{Good}}\left(t|c\right)=S_{\mathrm{Poor}}(t|c)$ against $H_{1}:S_{\mathrm{Good}}\left(t|c\right)\neq S_{\mathrm{Poor}}(t|c)$ for a large value of $\left|D\right|$, where
$$D=\frac{1}{\tau }\int_{0}^{\tau }\{{\hat{S}}_{\mathrm{Good}}^{\mathrm{CG}}\left(t|c,\alpha \right)-{\hat{S}}_{\mathrm{Poor}}^{\mathrm{CG}}(t|c,\alpha )\}dt,$$
where $\tau =\min\left\{\max_{{\;\mathrm{PI}}_{i}\leq c}\left(t_{i}\right),\max_{{\;\mathrm{PI}}_{i}>c}\left(t_{i}\right)\right\}$ is the largest time point where survivors are seen for both good and poor groups.

The *p*-value of the test is computed by $\sum_{r}1\left\{\left|D_{r}\right|>\left|D\right|\right\}/N$, where $D_{r}$ is a random permutation for D based on $\left\{(t_{r\left(i\right)},\;\delta_{r\left(i\right)},\;x_{i});\;i=1,\;2,\ldots ,\;n\right\}$ for a random permutation $r:\left\{1,\;2,\ldots ,\;n\right\}\to \;\left\{1,\;2,\ldots ,\;n\right\}$. To stabilize, we suggest a large number, say N=10,000. Note that τD is known as the restricted mean survival time difference (RMSTD).

The actual implementation of the proposed sensitivity analysis is technical. Thus, the development of the software implementation and the online web application are desirable for clinicians and patients to use. We leave this topic to Section 4. The following subsection provides specific examples of parametric copulas that are currently implemented in our developed R functions and web application.

### 3.2. Examples of Parametric Copulas

This subsection describes some examples of parametric copulas in order to perform a sensitivity analysis.

If the generator is $\phi_{\alpha }\left(t\right)=\left(t^{-\alpha }-1\right)/\alpha$ for α>0, it yields the Clayton copula
$$\phi_{\alpha }^{-1}\left[\phi_{\alpha }\left(v\right)+\phi_{\alpha }\left(w\right)\}\right]={\left(v^{-\alpha }+w^{-\alpha }-1\right)}^{-\frac{1}{\alpha }},\;0<v,w<1,\;\alpha >0$$

Kendall’s tau for measuring dependency is given by $\alpha /\left(\alpha +2\right)$. The resultant Clayton CG estimator is
$${\hat{S}}_{\mathrm{Good}}^{\mathrm{CG}}\left(t|c,\alpha \right)={\left[1-\underset{i:t_{i}\leq t,\;\delta_{i}=1,{\;\mathrm{PI}}_{i}\leq c}{\sum }\left\{{\left(\frac{{\overset{‾}{Y}}_{\mathrm{Good}}\left(t_{i}\right)-1}{n_{\mathrm{Good}}}\right)}^{-\alpha }-{\left(\frac{{\overset{‾}{Y}}_{\mathrm{Good}}\left(t_{i}\right)}{n_{\mathrm{Good}}}\right)}^{-\alpha }\right\}\right]}^{-\frac{1}{\alpha }}.$$

If the generator is $\phi_{\alpha }\left(t\right)={\left(-\log t\right)}^{\alpha +1}$ for α≥0, it yields the Gumbel copula
$$\phi_{\alpha }^{-1}\left[\phi_{\alpha }\left(v\right)+\phi_{\alpha }\left(w\right)\}\right]=\exp\left(-{\left[{\left(-\log v\right)}^{\alpha +1}+{\left(-\log w\right)}^{\alpha +1}\right]}^{\frac{1}{\alpha +1}}\right),\;0<v,w<1,\;\alpha \geq 0$$

Kendall’s tau for measuring dependency is given by $\alpha /\left(\alpha +1\right)$. The resultant Gumbel CG estimator is
$${\hat{S}}_{\mathrm{Good}}^{\mathrm{CG}}(t|c,\alpha )=\exp\left(-{\left[\underset{i:t_{i}\leq t,\;\delta_{i}=1,{\;\mathrm{PI}}_{i}\leq c}{\sum }\left\{{\left(-\log\left(\frac{{\overset{‾}{Y}}_{\mathrm{Good}}\left(t_{i}\right)-1}{n_{\mathrm{Good}}}\right)\right)}^{\alpha +1}-{\left(-\log\left(\frac{{\overset{‾}{Y}}_{\mathrm{Good}}\left(t_{i}\right)}{n_{\mathrm{Good}}}\right)\right)}^{\alpha +1}\right\}\right]}^{\frac{1}{1+\alpha }}\right).$$

Finally, one can obtain the Frank CG estimator by the generator $\phi_{\alpha }\left(t\right)=-\log\{\left(e^{-\alpha t}-1\right)/\left(e^{-\alpha }-1\right)\}$, α≠0.

By letting α=0 for the Gumbel copula, one has $\phi_{\alpha =0}\left(t\right)=-\log\left(t\right)$, and hence, the resultant CG estimators reduce to the KM estimator, namely, ${\hat{S}}_{\mathrm{Good}}^{\mathrm{CG}}(t|c,0)={\hat{S}}_{\mathrm{Good}}^{\mathrm{KM}}(t|c)$. For the Clayton copula, one has $\phi_{\alpha }\left(t\right)=\left(t^{-\alpha }-1\right)/\alpha \to -\log\left(t\right)$ with α→0. Hence, ${\hat{S}}_{\mathrm{Good}}^{\mathrm{CG}}\left(t|c,\alpha \right)\to {\hat{S}}_{\mathrm{Good}}^{\mathrm{KM}}(t|c)$ as α→0. The case of the Frank copula is similar.

## 4. Software and Web Application

This section describes the development of software to establish a Shiny-based web application for the practical implementation of the proposed sensitivity analysis.

In order to implement the aforementioned sensitivity analyses (Section 3.1), we developed computational/graphical facilities for the CG estimators. First, we refined the computational routines for the CG estimators in the R package *compound.Cox* [9]. The package now includes the functions CG.Clayton(.), CG.Frank(.), and CG.Gumbel(.), which can compute the CG estimators under the three copulas. These functions compute the CG estimators and graphically display the plot possibly colored by an option (the *S.col* option). This is useful for visually distinguishing the good prognosis group (specify *S.col=“blue”*) and poor prognosis group (specify *S.col=“red”*). Second, we devise a new function, CG.test(.), for testing the prognostic difference for the good and poor prognosis groups based on the difference statistic between two CG estimates. The *p*-value is computed by the permutation test (Section 3.1). This is useful for objectively measuring the prognostic ability of the PI. While N=10,000 is recommended, this number may be reduced to N=200 to produce a web application that works reasonably fast.

An example of the CG estimator under the Clayton copula is given in Figure 2. Users may run the R code (upper-left panel) after installing the *compound.Cox* package. Then, the plot of the CG estimates is displayed in the right panel. The numerical values of the estimates (survival probabilities at $t_{i}$) are shown in the lower-left panel. This example uses data $t_{i}=$1, 3, 5, 4, 7, 8, 10, 13; $\delta_{i}=$1, 0, 0, 1, 0, 1, 0. The data are artificial data, which may be regarded as patients from a good prognosis group (hence, we specify the *S.col =“blue”* option to make the plot blue-colored). The Clayton copula’s parameter is specified as α=18 (Kendall’s tau is 0.9). If one specifies the copula parameter as α=0, this yields the KM estimator. We see that the numerical results are identical to the results computed by the *survfit(.)* function (lower-left panel).

An example for testing the difference for the good and poor prognosis is given in Figure 3. This example uses prognostic indices ${\mathrm{PI}}_{i}=$ 8, 7, 6, 5, 4, 3, 2, 1 to predict the survival outcomes $t_{i}$ and $\delta_{i}$ already defined. The figure displays the plots of the Clayton CG estimates that are separated between good and poor prognoses group. In addition, the figure shows the *p*-value computed by the permutation test. More detailed results, such as the difference (D) and the RMSTD (τD), are given in the left panel.

While we developed the infrastructures of the computational/graphical facilities for the CG estimators, the actual implementation of the sensitivity analysis is still complex. This is because the sensitivity analysis requires the understanding of the R programming and the proposed methods of Section 3. To further make our methodologies accessible to users, we propose a method to produce an interactive web application that can be manipulated by users with minimal knowledge (e.g., clinicians and patients).

We suggest an R package *Shiny* to make an online web application that can display the plots of ${\hat{S}}_{\mathrm{Good}}^{\mathrm{CG}}(t|c,\alpha )$ and ${\hat{S}}_{\mathrm{Poor}}^{\mathrm{CG}}(t|c,\alpha )$ given a user-specified form of $\phi_{\alpha }\left(.\right)$ and a value of α. With *Shiny*, one can easily convert R programs into a web application. This process can be carried out by creating the “app.R” file, a program file written in the R language. We develop a template file (available in Appendix A) so that developers can modify the template for their own prediction settings. Once the template is appropriately tuned by developers, the web application is immediately built and can be made publicly available through a platform *shinyapps.io* (https://www.shinyapps.io/ (accessed on 10 January 2023). A concrete implementation of the web application shall be described in the next section.

## 5. Results

The purpose of this section is to demonstrate the proposed methods of Section 3 by using the lung cancer data originally reported in [6] and now available in the *Lung* object in the *compound.Cox* package [9]. We first introduce the dataset.

### 5.1. Lung Cancer Data

Briefly, the *Lung* data are a survival dataset containing n=63 training samples and n=62 test samples of 125 surgically treated non-small-cell lung cancer patients [6]. The endpoint of interest is overall survival { XE “overall survival:overall ”}, i.e., months from surgery to death. Overall survival time and censoring status (death or alive) were recorded. Additionally, available are high-dimensional gene expressions that may predict survival. All the gene expressions were coded as 1, 2, 3, or 4 according to the quartile levels following the original study [6].

In the following analysis, we selected the 16 genes and then constructed the multi-gene predictor as described in Section 2.2–Section 2.3 (also reported in [37,41]). The resultant predictor takes the form ${\mathrm{PI}}_{i}\left(\hat{\alpha }\right)={\hat{\beta }}_{1}\left(\hat{\alpha }\right)x_{i,1}+\;\cdots +\;{\hat{\beta }}_{16}\left(\hat{\alpha }\right)x_{i,16}$ for the *i*-th patient in the test samples, where ${\hat{\beta }}_{j}\left(\hat{\alpha }\right)$ is computed by the training samples given the copula parameter $\hat{\alpha }=18$ (Kendall’s tau = 0.90), where $\left(x_{1},\;\cdots ,\;x_{16}\right)$ are gene expression{ XE “gene expression:gene expression ”}s taking values 1, 2, 3, and 4. In terms of gene symbols, the 16-gene predictor is
PI = (0.51 × MMP16) + (0.51 × ZNF264) + (0.50 × HGF) + (−0.49 × HCK) + (0.47 × NF1) + (0.46 × ERBB3) + (0.57 × NR2F6) + (0.77 × AXL) + (0.51 × CDC23) + (0.92 × DLG2) + (−0.34 × IGF2) + (0.54 × RBBP6) + (0.51 × COX11) + (0.40 × DUSP6) + (−0.37 × ENG) + (−0.41 × IHPK1).


In order to examine the sensitivity of the PI under a variety of dependent censoring scenarios, we shall apply the proposed sensitivity analysis methods.

### 5.2. Sensitivity Analysis via the Web App

As prescribed in Section 3, we formed a good prognosis group $\left\{i;{\mathrm{PI}}_{i}\left(\hat{\alpha }\right)\leq c\right\}$ and a poor prognosis group $\left\{i;{\mathrm{PI}}_{i}\left(\hat{\alpha }\right)>c\right\}$, where the cut-off value c was set to be the 50th percentile (median) of the PIs in the training samples (as we proposed in Section 3). The test samples were then separated into two groups. Then, we evaluated the prognostic performance of the PI by displaying the estimated survival functions ${\hat{S}}_{\mathrm{Good}}^{\mathrm{CG}}(t|c,\alpha )$ and ${\hat{S}}_{\mathrm{Poor}}^{\mathrm{CG}}(t|c,\alpha )$ given a copula and a value of α.

We constructed a web application (Figure 4) by the methods of Section 4. The interactive web version is available in https://takeshi.shinyapps.io/lung/ (accessed on 10 January 2023). The main goal of this web application is to visualize how the two prognosis groups are separated by the PI for a user-specified choice of copulas and any value of α.

Using this web application, we tried the Clayton, Gumbel, and Frank copulas under several values of α. For the Clayton copula and α=2, one can see a good separation between the plots of ${\hat{S}}_{\mathrm{Good}}^{\mathrm{CG}}(t|c,\alpha )$ and ${\hat{S}}_{\mathrm{Poor}}^{\mathrm{CG}}(t|c,\alpha )$ as shown in the right panel of Figure 4; one can also see the *p*-value (=0.04 based on *N* = 200 permutations).

Table 1 summarizes the *p*-values under the three copulas with selected values of α whose Kendall’s measures of correlation (tau) take −0.75, −0.46, 0, 0.33, 0.46, 0.50, and 0.75. The *p*-values were obtained based on *N* = 10,000 permutations (by setting the left panel of Figure 4). One can observe that the *p*-values obtained are less than or around 0.05 for positive dependence (Kendall’s tau = 0.75 or 0.50), while they are greater than 0.10 for negative dependence and independence (tau = −0.75, −0.46, 0). Therefore, the PI would lose the ability of prognostic prediction when censoring in the test data is negatively dependent on survival. Therefore, our analysis reveals a potential drawback for the PI developed by Emura and Chen [41] when negatively dependent censoring arises. On the other hand, the PI gains its improved prognostic ability when censoring is positively dependent on survival. The *p*-value reaches nearly 1% level for strongly positive dependence (tau = 0.88 or 0.91). This is reasonable since the PI is developed under the Clayton copula with strongly positive dependence ($\hat{\alpha }=18$, tau = 0.90).

Recall that the cut-off value c was set to be the median of ${\mathrm{PI}}_{i}\left(\hat{\alpha }\right)$’s. If we change the cut-off value to be the first or third quartile (25% or 75% points) of ${\mathrm{PI}}_{i}\left(\hat{\alpha }\right)$’s, then the separation between the plots of ${\hat{S}}_{\mathrm{Good}}^{\mathrm{CG}}(t|c,\alpha )$ and ${\hat{S}}_{\mathrm{Poor}}^{\mathrm{CG}}(t|c,\alpha )$ is not clear. This confirms our suggestion to apply the median as the cut-off value.

### 5.3. Comparison with the 16-Gene Pedictor Developed by Chen et al. [6]

The proposed sensitivity analysis is useful for any type of established predictor, not restricted to the predictor developed by the copula-based gene selection method. Furthermore, the proposed web application allows one to compare different predictors in a user-friendly manner. In order to demonstrate this feature, we performed a sensitivity analysis for the PI based on Chen et al. [6] defined as
PI = (−1.09 × ANXA5) + (1.32 × DLG2) + (0.55 × ZNF264) + (0.75 × DUSP6) + (0.59 × CPEB4) + (−0.84 × LCK) + (−0.58 × STAT1) + (0.65 × RNF4) + (0.52 × IRF4) + (0.58 × STAT2) + (0.51 × HGF) + (0.55 × ERBB3) + (0.47 × NF1) + (−0.77 × FRAP1) + (0.92 × MMD) + (0.52 × HMMR).

Table 1 shows the *p*-values testing the difference of survival prognosis (good vs. poor) made by the PI. We observe that the *p*-values are mostly less than 0.05 for the three copulas with selected values of α (Kendall’s tau takes −0.75, −0.46, 0, 0.33, 0.46, 0.50, 0.75). Therefore, the predictive ability of the PI is justified for a variety of dependent censoring models. This extremely confirms the validity and robustness of the PI developed by Chen et al. [6], even though they are developed under the independent censoring model. On the other hand, the ability of the PI under strong positive dependence is reduced compared to the PI developed by Emura and Chen [41]. If strong positive dependence exists in the test samples, the latter PI better prognosticates survival. However, we do not suggest determining the single best copula and the single value of α in the sensitivity analysis based on the CG estimators [52,53,54,55,56]. Instead, we suggest tying different copulas and their parameters to understand the sensitivity of the PI.

## 6. Conclusions and Discussion

In this article, we propose a novel sensitivity analysis method for a multi-gene predictor for survival. The primary tool for the proposed method is the CG estimator [52], which can effectively adjust for the impact of dependent censoring on survival data. Consequently, a unique feature of the proposed method is the ability to handle dependent censoring, which has been of increasing concern to medical statisticians in a variety of survival analysis methods [41,42,43,44,45,46,47,48,49,50,51,52,53,54,55,56]. The lung cancer data analysis demonstrates the implementation of the proposed method on a web application (Figure 4; https://takeshi.shinyapps.io/lung/ (accessed on 10 January 2023)). A user of this application can easily perform a sensitivity analysis on survival prognostic models using a smartphone.

From a methodological point of view, the proposed method is a novel implementation of the CG estimator [52] to the context of prognostic prediction. So far, the CG estimator has been applied to a single population setting [44,51,52,53,54], two-sample comparison [55], simple regression [43], multiple regression [60,61], factorial designs [56], and survival forests [47]. The objective of survival forests is to classify patients into several groups, which is similar to the goal of the proposed method. The difference is that survival forests use a decision tree while ours uses a multi-gene predictor. However, survival forests have not been adapted to deal with high-dimensional gene expressions. Hence, it is of great interest to extend survival forests to handle both dependent censoring and high-dimensional covariates as in a survival tree [62].

In this article, we employed the R package *Shiny* to transfer our sensitivity analysis formulas to a user-friendly web application. Some authors have used the Shiny package in other contexts. A series of works by Fournier et al. [63], Asar et al. [64], and Lenain et al. [65] made a Shiny-based web application for the dynamic prediction of long-term kidney graft failure. Similarly, the dynamic prediction method of Emura et al. [21] (see also the updated works of [66]) was transferred to web applications [67]. Clearly, this type of survival prognostic prediction tool can promote the development of personalized medicine, allowing clinicians to utilize powerful prognostic prediction models.

We note that the issue of dependent censoring in survival prognostic prediction is not restricted to medical research, which can be seen in the reliability analysis of mechanical items or systems [68,69,70,71,72,73]. In the reliability prediction of mechanical equipment, the survival probability is usually modeled by parametric distributions, such as the Weibull distribution. The analysis of a prognostic prediction model in engineering settings poses additional challenges tailored to parametric analyses. Recall that the CG estimator is a semiparametric estimator that does not assume any parametric form of the survival function. The semiparametric CG estimator [52] is not easily modified to parametric models since the estimator may require some numerical integrations.

We focus on the three Archimedean copulas: the Clayton, Gumbel, and Frank copulas, which have been extensively used in survival analysis [74,75,76,77,78] and other fields [79,80,81]. Owing to their remarkable popularity, our choices of copulas are natural. Furthermore, these Archimedean copulas allow simple formulas to compute the CG estimators [52]. However, there are a large number of non-Archimedean copulas popular in a variety of applications, such as the FGM copula, Gaussian copula, trigonometric copula, and Celebioglu–Cuadras copula [82,83,84,85,86,87]. As the CG estimators have not been considered for these non-Archimedean copulas, it is of great interest to develop computational tools for them.

## Figures and Tables

**Figure 1 biomedicines-11-00797-f001:**
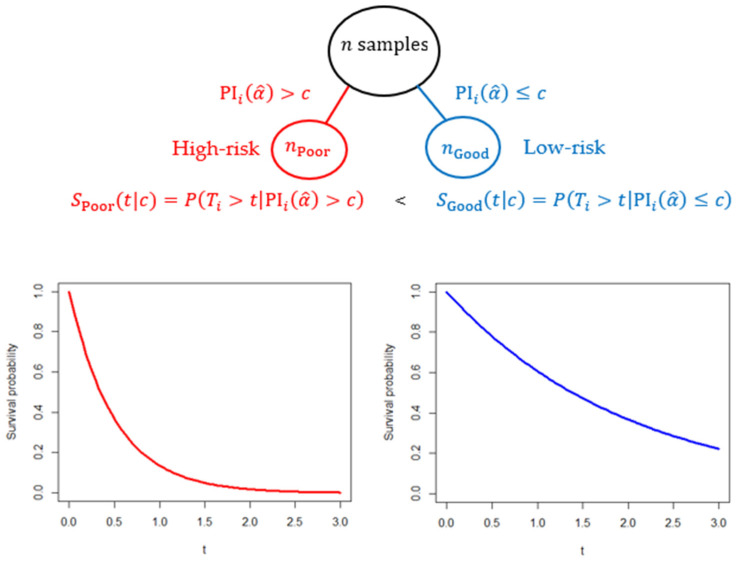
Good and poor prognosis group separated by the PI applied for the testing sample of size *n*. Concretely, $\left\{i;{\mathrm{PI}}_{i}\left(\hat{\alpha }\right)\leq c\right\}$ is a good prognosis (low-risk) group, and $\left\{i;{\mathrm{PI}}_{i}\left(\hat{\alpha }\right)>c\right\}$ is a poor prognosis (high-risk) group. If the classification is successful, survival functions exhibit $S_{\mathrm{Good}}\left(t|c\right)>S_{\mathrm{Poor}}(t|c)$. The cut-off value c can be the median of the PI values calculated from the training samples.

**Figure 2 biomedicines-11-00797-f002:**
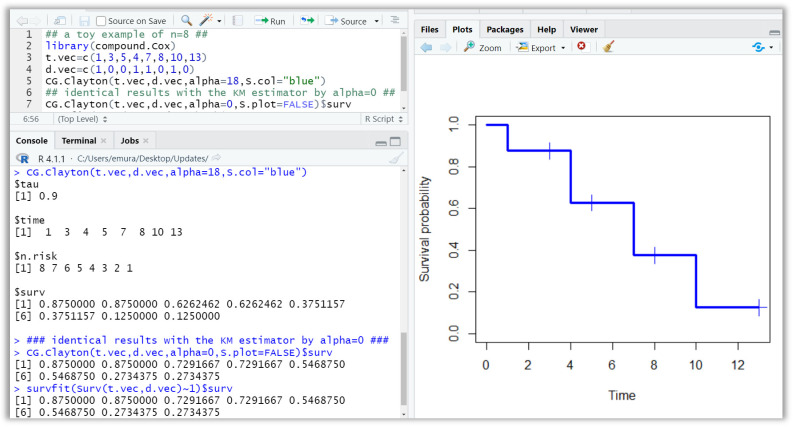
The R console showing an example of the CG estimator under the Clayton copula. The upper-left panel is the R code. The right panel is the plot of the Clayton CG estimates computed by data $t_{i}=$ 1, 3, 5, 4, 7, 8, 10, 13; $\delta_{i}=$ 1, 0, 0, 1, 0, 1, 0. The lower-left panel shows the numerical values of the estimates (survival probabilities). The Clayton copula’s parameter is specified as α=18 (Kendall’s tau is 0.9). If one specifies the copula parameter as α=0, this yields the KM estimator. We see that the numerical results are identical to the results computed by the *survfit(.)* function (lower-left panel).

**Figure 3 biomedicines-11-00797-f003:**
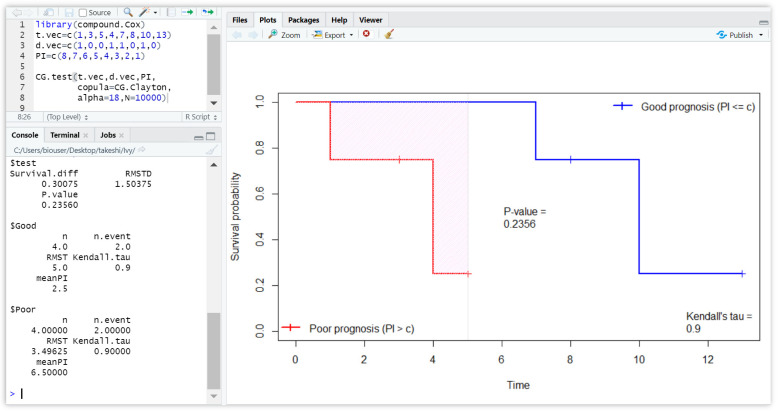
The R console showing the test results for the difference for the good and poor prognoses. The right panel gives the plots of the Clayton CG estimates computed by data $t_{i}=$ 1, 3, 5, 4, 7, 8, 10, 13; $\delta_{i}=$ 1, 0, 0, 1, 0, 1, 0; ${\mathrm{PI}}_{i}=$ 8, 7, 6, 5, 4, 3, 2, 1. The Clayton copula’s parameter is specified as α=18. More detailed results, such as the difference (D) and the RMSTD (τD), are given in the left panel.

**Figure 4 biomedicines-11-00797-f004:**
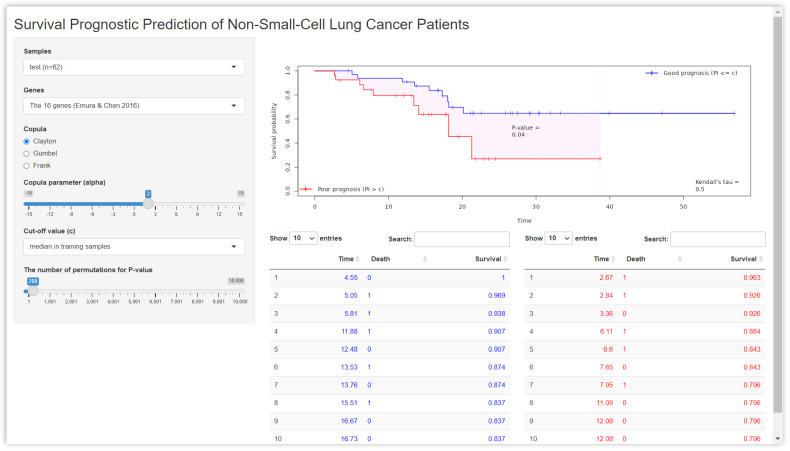
The web application for the sensitivity analysis based on the lung cancer data. The interactive web version is available in https://takeshi.shinyapps.io/lung/ (accessed on 10 January 2023). The main goal of this web application is to visualize how the two prognosis groups are separated by the 16-gene PI for a user-specified choice of $\phi_{\alpha }\left(.\right)$ (copula) and any value of α (copula parameter).

**Table 1 biomedicines-11-00797-t001:** *p*-values for testing the difference of survival prognosis (good vs. poor) under the three copulas with selected values of α (Kendall’s tau takes −0.75, −0.46, 0, 0.33, 0.46, 0.50, 0.75).

		Prognostic Index (PI)	
		Emura and Chen [41]	Chen et al. [6]
Clayton copula	α=1 (tau = 0.33)	0.065	0.036
	α=2 (tau = 0.50)	0.036	0.036
	α=4 (tau = 0.75)	0.021	0.036
	α=15 (tau = 0.88)	0.011	0.029
Gumbel copula	α=0 (tau = 0.00)	0.171	0.040
	α=1 (tau = 0.50)	0.058	0.029
	α=2 (tau = 0.75)	0.028	0.029
	α=10 (tau = 0.91)	0.011	0.025
Frank copula	α=−14 (tau = −0.75)	0.360	0.059
	α=−5 (tau = −0.46)	0.307	0.054
	α=5 (tau = 0.46)	0.054	0.034
	α=14 (tau = 0.75)	0.018	0.032

**Notes:** The PI is either 16-gene PI of Emura and Chen [41] or the 16-gene PI of Chen et al. [6]. The *p*-value is obtained by *N* = 10,000 permutations.

## Data Availability

All the results in this article are reproducible by the code available in Appendix A.

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
