# Peer review of "Sensitivity Analysis for Survival Prognostic Prediction with Gene Selection: A Copula Method for Dependent Censoring"

_biomedicines, 2023, doi:10.3390/biomedicines11030797_

Round 1

Reviewer 1 Report

In this manuscript, the authors proposed to use the copula-graphic estimator to select genes and build the predictive model using gene expression data. There are two main contributions of this manuscript: First, the authors implemented a R package and a web application so the researchers can use the methods described in the manuscript. Second, the authors evaluated the performance (sensitivity) of copula-graphic estimator to select genes and predict survival time using gene expression data. The manuscript is well written and the methods used are well described. However, the copula method that considers the dependent censoring has been proposed to select genes that can be used for survival prediction using gene expression data. The authors performed a simple study to evaluate how well such selection based on the cupola method performs. I do not think the manuscript is highly innovation.

One of the major contributions that the authors stated is to use the copula method to select genes, build predictive model, and calculate survival probabilities. Specifically, the authors proposed: (1) Use the copula method to estimate regression coefficients of genes and select genes with small p-value; (2) Use selected genes and estimated regression coefficients to build a liner predictor; and (3) calculate survival probabilities using the liner predictor with different copula method. All methods described in the manuscript have been developed before, so the innovation is not high. In addition, additional work is needed for their sensitivity analysis.

Lines 272-273: The process should be repeated multiple times.

Line 291-292: It seems that c is selected based on the median of PI from the testing data. Why the median is used? In addition, such selection of c should be based on the training data. Although the authors mentioned the other values of c. It would great if the authors can describe a systematic way to select c based on the training data.

Lines 294-295, the evaluation is based on the visual inspection of plots. It may be subjective and may not be used to compare the performance of different methods.

Different methods (C-CG, G-Cg, F-CG) and values of parameter in the copula can be used. Which one is better? If different results are obtained based on different methods and values of parameter, which results can be used and how results should be interpreted.

A comparison study between methods based on classical Cox regression and methods based on copula graphic estimators should be added.

Author Response

Many thanks for your comments. Please see the attached file for our point-by-point responses.

Reviewer 2 Report

The presented article is quite interesting. The article has both medical and mathematical aspects. I can't comment on the medical side because I'm just a reader in this area. However, the mathematical side of the article should be written clearly and comprehensibly.

Below are some notes.

   (1) Actually survival function is the conditional survival function. The same can be said for all the other functions used.

   (2) Line 75, Line 83, Line 120. Use of conditions x_i and x_ij must be coordinated. There cannot be x_i on one side of the equation and x_ij on the other.

    (3) Lines 112-114 . The names of the copula variables need to be unified.

    (4) Beginning of section 3. The value of the constant c obtained from is not clear. It is not at all clear what Figure 1 shows.

    (5) Section 3.1. Where does the parameter theta come from? Is alpha the same dependency parameter here? Is there another one? In any case, section 3.1 needs to be fundamentally rewritten. This section is the most important, but the worst written.

Author Response

(The authors gave the same response as above.)

Reviewer 3 Report

The following articles should be consulted.

10.1016/j.gene.2022.146961

10.12114/j.issn.1007-9572.2022.0327

10.21037/jgo-22-1134

10.1097/MD.0000000000032558

10.3389/fonc.2022.1030802

10.3389/fimmu.2022.1110602

10.3389/fimmu.2022.1076045

10.3389/fonc.2022.1056623

10.3389/fonc.2022.983956

10.1182/hematology.2022000339

10.1016/j.intimp.2022.109335

10.1038/s41598-022-23852-z

10.1186/s12920-022-01339-0

10.1186/s12920-022-01341-6

Author Response

(The authors gave the same response as above.)

Round 2

Reviewer 1 Report

The authors have responsively addressed comments from two reviewers so I do not have further comments.

Reviewer 2 Report

The mathematical part has been corrected. I have no more complaints.